# Live birth in an archosauromorph reptile

Jun Liu[1,2,3], Chris L. Organ[4], Michael J. Benton[5], Matthew C. Brandley[6] & Jonathan C. Aitchison[7]

Live birth has evolved many times independently in vertebrates, such as mammals and diverse groups of lizards and snakes. However, live birth is unknown in the major clade Archosauromorpha, a group that first evolved some 260 million years ago and is represented today by birds and crocodilians. Here we report the discovery of a pregnant long-necked marine reptile (*Dinocephalosaurus*) from the Middle Triassic (~245 million years ago) of southwest China showing live birth in archosauromorphs. Our discovery pushes back evidence of reproductive biology in the clade by roughly 50 million years, and shows that there is no fundamental reason that archosauromorphs could not achieve live birth. Our phylogenetic models indicate that *Dinocephalosaurus* determined the sex of their offspring by sex chromosomes rather than by environmental temperature like crocodilians. Our results provide crucial evidence for genotypic sex determination facilitating land-water transitions in amniotes.

[1] School of Resources and Environmental Engineering, Hefei University of Technology, Hefei 230009, China. [2] Chengdu Center, China Geological Survey, Chengdu 610081, China. [3] State Key Laboratory of Palaeobiology and Stratigraphy, Nanjing Institute of Geology and Palaeontology, CAS, Nanjing 210008, China. [4] Department of Earth Sciences, Montana State University, Bozeman, Montana 59717, USA. [5] School of Earth Sciences, University of Bristol, Bristol BS8 1RJ, UK. [6] School of Life and Environmental Sciences, The University of Sydney, Sydney, New South Wales 2006, Australia. [7] School of Earth and Environmental Sciences, University of Queensland, Brisbane, Queensland 4072, Australia. Correspondence and requests for materials should be addressed to J.L. (email: junliu@hfut.edu.cn).

Adaptations related to reproduction directly affect the ability of organisms to produce subsequent generations and persist over evolutionary time scales[1]. The evolutionary transition from egg-laying (oviparity) to live birth (viviparity) involves subtle changes to maternal morphology, physiology, and behaviour, which can lead to matrotrophy (feeding by the mother, for example through a placenta) and shifts in ecological and evolutionary trajectories[1–3]. Despite the complexity of this transition, viviparity has evolved at least 115 times in extant squamates (lizards and snakes), in addition to a single time in the common ancestor of therian mammals[2,4]. Moreover, viviparity is a common reproductive mode in extinct aquatic reptiles[5] including eosauropterygians[6,7], ichthyosaurs[8–12], mosasauroids[13,14], some choristoderans[15] and likely mesosaurs[16,17]. However, all of the above viviparous reptile lineages are concentrated within one of the three major lineages of extant reptiles—the Lepidosauromorpha—plus some completely extinct aquatic groups with uncertain affinities (Fig. 1). No evidence for viviparity has been discovered in the two other major lineages, Testudines (turtles) and Archosauromorpha.

Basal archosauromorph reptiles first appeared in the Late Permian and diversified in the Triassic[18]. They include trilophosaurs, protorosaurs, rhynchosaurs and archosauriforms, the latter of which includes the ancestors of the crown-group, namely the birds and crocodilians, and their extinct relatives including non-avian dinosaurs and pterosaurs among others[18]. All crown-group archosauromorphs lay calcified eggs[19], the fossil record of which can be traced back at least to the Early Jurassic[20,21]. However, the reproductive biology of stem-group archosauromorphs remains unknown.

Here we report a new specimen of the aquatic protorosaur *Dinocephalosaurus*[22] from the Middle Triassic of South China containing an embryo in the abdominal region. The pregnant specimen provides evidence of live birth in a reptile with undoubted archosauromorph affinity and insight into the reproductive biology of stem-group archosauromorphs.

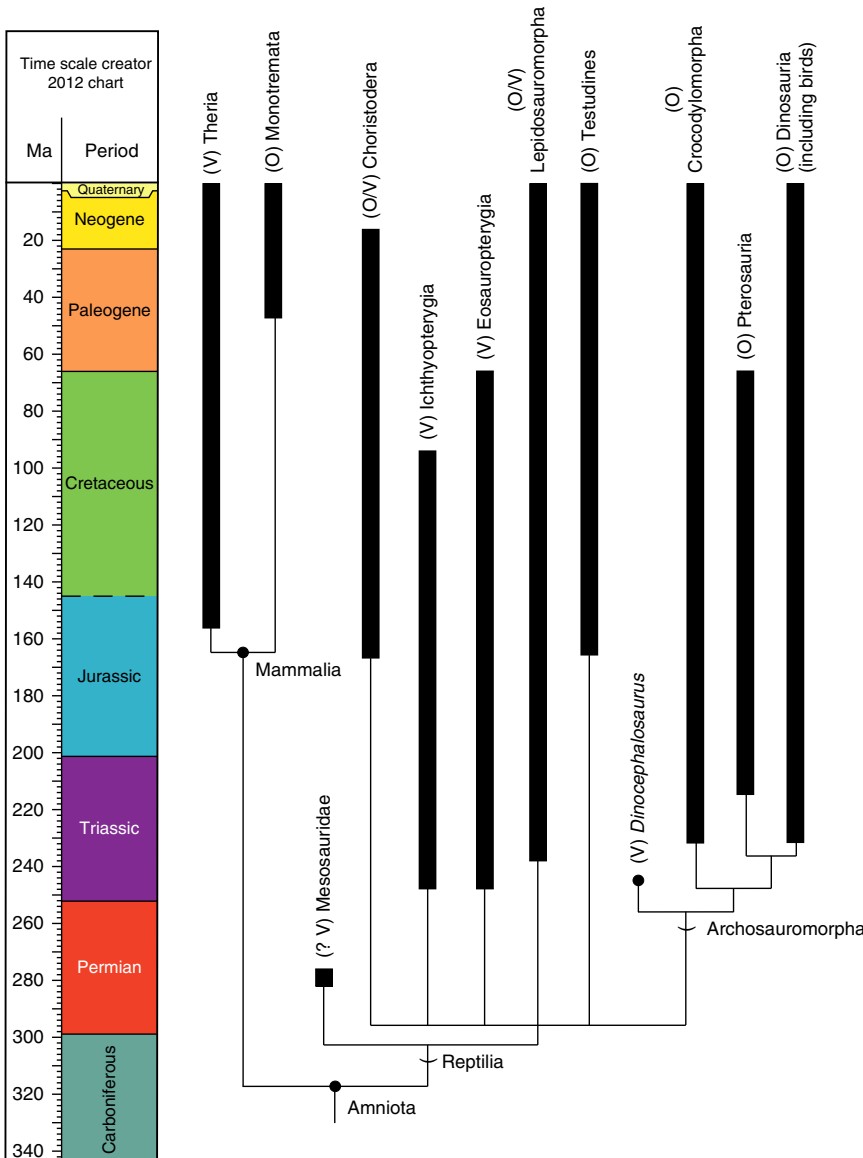

**Figure 1 | Evolution of reproductive modes in major groups of amniotes.** The phylogenetic tree is derived from a combination of published sources[18,45]. O, egg-laying (oviparous); V, live-bearing (viviparous).

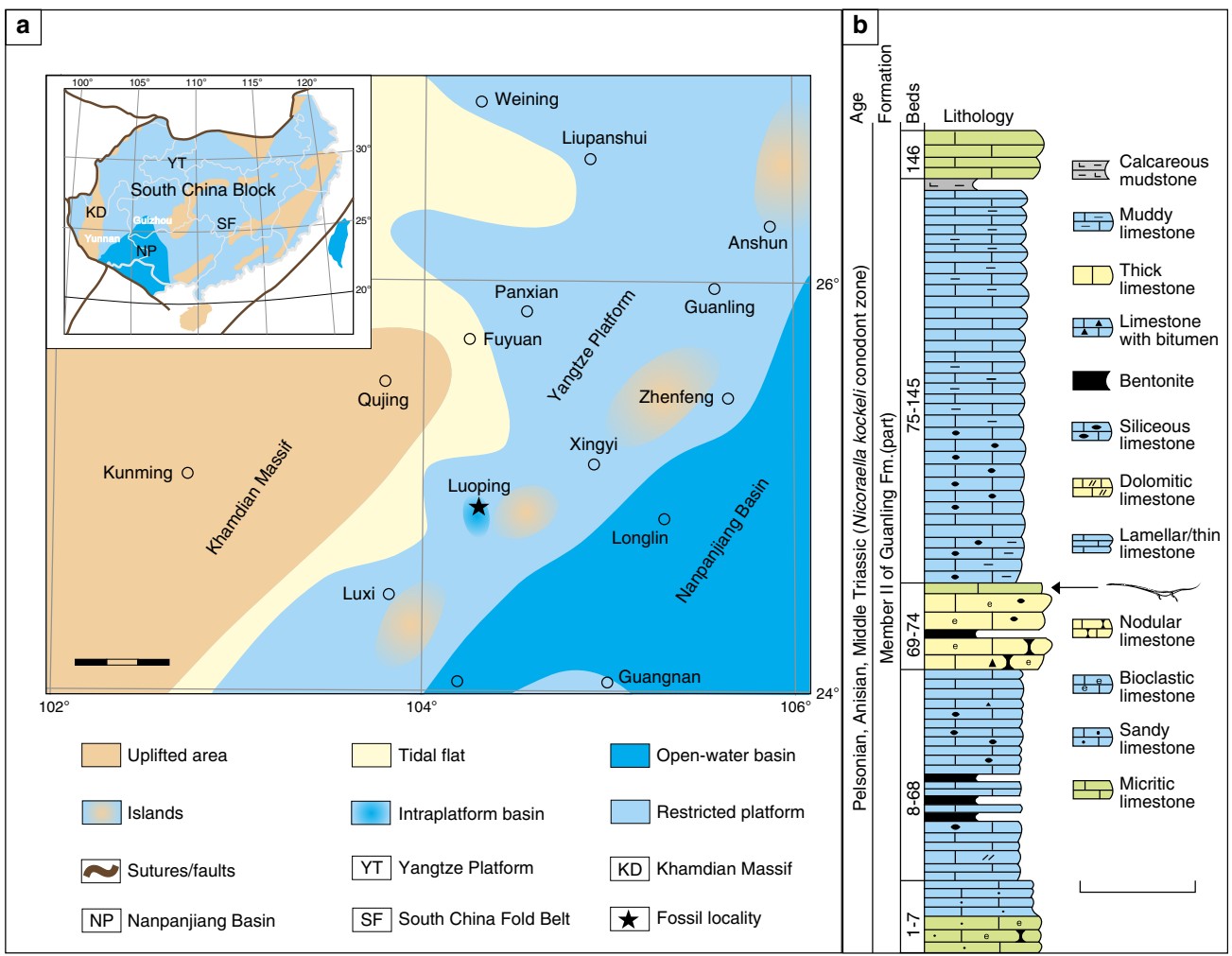

**Figure 2 | Locality and horizon of the new *Dinocephalosaurus* specimen LPV 30280.** (**a**) Paleotectonic map showing the location of the Luoping biota where LPV 30280 is preserved[23]. Scale bar, 60 km. (**b**) Stratigraphic column of the Dawazi section[23] showing the horizon of LPV 30280. Scale bar, 2 m.

## Results

**Geological background.** The new specimen belongs to the recently discovered Luoping biota[23,24], which is preserved in an intraplatform basin of the Yangtze Platform in the South China Block (Fig. 2a). Conodont analysis dates the biota to the Pelsonian substage of the Anisian in the Middle Triassic[25], corresponding to an age of ∼244–245 million years ago (Ma). The Luoping biota comprises thousands of extremely well preserved fossil specimens[23,24]. The specimen was collected in 2008 from Bed 74 of the Dawazi section (Fig. 2b) in Luoping County, Yunnan Province, China. The thin micritic limestone layer bearing the specimen immediately overlies a set of thick-bedded siliceous nodular limestones (Fig. 2b). Before collection, the specimen had already been weathered into three blocks in the field, the fractures of which were filled with modern soils. The specimen was then transferred from Luoping to the Chengdu Center of China Geological Survey for routine mechanical preparation.

**Systematic palaeontology and phylogenetic analysis.** The Luoping specimen (Fig. 3) is catalogued as LPV 30280 in the Chengdu Center of China Geological Survey. It shares several unequivocally derived characters with published *Dinocephalosaurus* specimens among protorosaurs, including

a remarkably elongated neck formed by an increased number of cervical vertebrate, free sacral and caudal ribs from relevant vertebrae, round tarsus ossifications and moderate hyperphalangy in the pes. These distinct synapomorphies make the identification of the new material as *Dinocephalosaurus* unambiguous. Slight differences exist, but currently the available specimens are not enough to reach a firm conclusion that these differences are stable between specimens from two different locations (see Supplementary Note 1 for detailed description of the specimen). Using both parsimony and Bayesian methods, phylogenetic analyses incorporating all well-known protorosaurian genera reveal a closer phylogenetic relationship of *Dinocephalosaurus* to the derived tanystropheids than to other protorosaurs (Fig. 4; Supplementary Figs 1 and 2).

**Description of the embryo.** The embryo preserved in specimen LPV 30280 contains several pieces of mostly articulated cervical vertebrae associated with cervical ribs, part of the forelimbs and pieces of other unidentifiable elements (Fig. 3c,d). It clearly belongs to the same species as the adult, as indicated by the shared morphology, including greatly elongated cervical vertebrae, very low cervical neural spines and elongated cervical ribs extending across at least three intervertebral articulations. The identification of this embryo as *Dinocephalosaurus* is

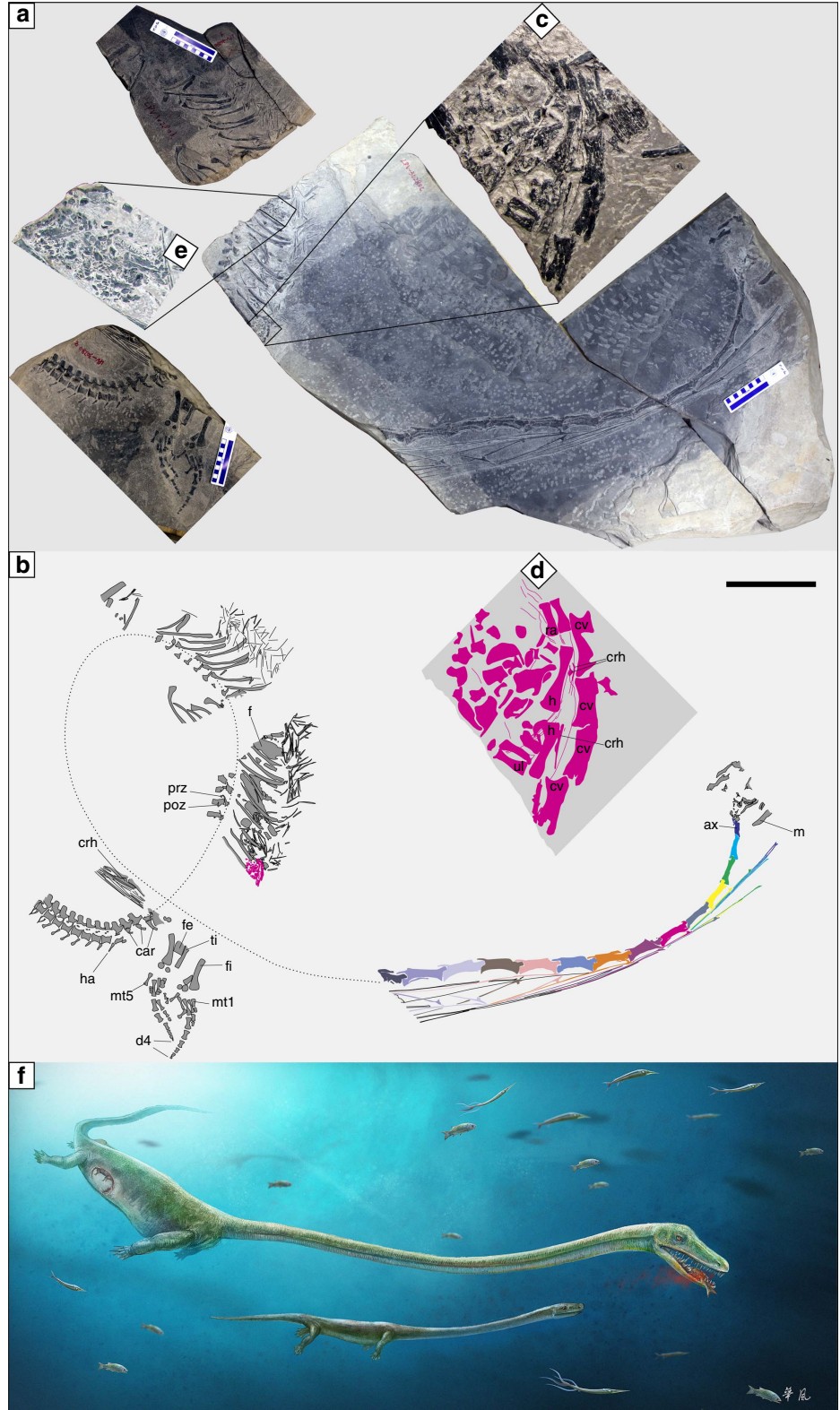

**Figure 3 | Skeleton of the new *Dinocephalosaurus* specimen LPV 30280.** (**a**) Photograph. The three separate blocks are arranged following their original positions in the field. (**b**) Interpretive drawing. Dotted line indicates the rough course of the vertebral column of the adult. The different colour in the cervical region aims to facilitate the association of cervical ribs with corresponding vertebrae. (**c**) Photo showing a close-up of the embryo preserved in the stomach region of LPV 30280. (**d**) Interpretive drawing of the embryo. (**e**) Photo showing a close-up of the perleidid fish preserved in the stomach region of LPV 30280. (**f**) Artist's reconstruction of *Dinocephalosaurus* showing the rough position of the embryo within the mother. ax, axis; car, caudal rib; crh, cervical rib head; cv, cervical vertebrae; d4, fourth digit; f, perleidid fish; fe, femur; fi, fibula; h, humerus; ha, haemal arch; m, mandible; mt1, metatarsal 1; mt5, metatarsal 5; poz, postzygapophysis; prz, prezygapophysis; ra, radius; ti, tibia; ul, ulna. Scale bar, 20 cm.

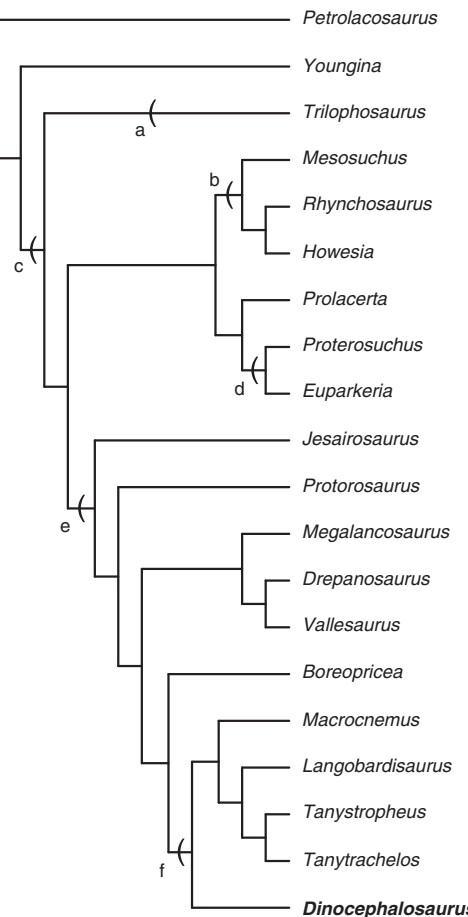

**Figure 4 | Phylogenetic hypothesis of *Dinocephalosaurus*.** Key clades are indicated: a Trilophosauria, b Rhynchosauria, c Archosauromorpha, d Archosauriformes, e Protorosauria and f Tanystropheidae.

further supported by the absence of any protorosaur other than *Dinocephalosaurus* in the Anisian of South China. Both forelimbs are preserved, but the autopodia are absent. They may be unossified, as consistent with the general ossification sequence of tetrapod limbs[18], although the possibility of postmortem disarticulation and loss cannot be excluded. The humerus of the embryo is 15 mm long. In the only published adult *Dinocephalosaurus* specimen preserving a humerus, this bone is 121 mm long[26]. The maternal specimen preserves a complete 75 mm long fibula, which is the same size as in the published adult *Dinocephalosaurus* specimen[26]. Thus, by extrapolation the embryo is 12% of the body size of its mother.

## Discussion

Several lines of evidence identify the small *Dinocephalosaurus* specimen in the abdominal region of LPV 30280 as an embryo of the maternal specimen. First, the embryo is completely enclosed within the body of the maternal specimen, and this excludes the possibility of superposition. The second line of evidence comes from the orientation of the embryo, with the neck pointing forward. This is evidenced by the fact that the cervical rib head of the embryo and the prezygapophyses of the dorsal vertebrae of the adult point in the same direction (Fig. 3). In aquatic amniotes, prey is usually swallowed head-first and this orientation is maintained during digestion and disarticulation[13]. Indeed, the partially digested perleidid fish (Fig. 3e) preserved in the abdominal region of LPV 30280 is oriented in the head-backward

position. Therefore, the neck-forward position of the embryonic skeleton suggests that the included skeleton was not ingested prey, but was an embryo. Finally, we note that the embryo demonstrates the curled posture typical for vertebrate embryos. The neck of the embryo slightly curves towards the forelimbs and ribs (Fig. 3c,d), while in all other known adult specimens of *Dinocephalosaurus*, the neck curves towards the dorsal side[26].

In all archosauromorphs and turtles, the eggshells are well calcified, though many turtle eggs are also pliable[27]. There is no trace of preserved eggshells near the LPV 30280 embryo, while many delicate calcareous fossils are preserved in the same horizon[23]. This is consistent with the eggshell morphology of extant viviparous reptiles. Although an eggshell membrane initially forms around the developing embryo in viviparous reptile species, it does not become calcified[28]. Altogether, these lines of evidence suggest that the embryo was likely contained in soft, uncalcified membranes, as in living viviparous reptiles, although the taphonomic absence of a calcified shell cannot be excluded.

The second line of evidence for viviparity in LPV 30280 is that the bones of the embryo are well ossified, indicating a relatively advanced embryonic stage. Living archosauromorphs, crocodilians and birds, all lay eggs very early in embryonic development, the neurulation and blastulation stages, respectively[29]. The tuatara and turtles lay eggs even earlier, in the gastrula stage[29]. Thus, it is unlikely that oviparous protorosaurs, which are archosauromorphs (Fig. 4), laid eggs with embryos in late developmental stages.

Our conclusion that *Dinocephalosaurus* had live birth is consistent with its functional morphology[22]. The anatomy of *Dinocephalosaurus* demonstrates that it was a fully marine reptile, representing the climax of aquatic adaptation of protorosaurs[22]. This is supported by the presence of hyperphalangy in the LPV 30280 specimen (Fig. 3). Further adaptations to marine living include the large paddle-like limbs and extremely elongated neck, both of which make it unlikely that *Dinocephalosaurus* could function comfortably on land or easily build terrestrial nests similar to those of sea turtles. Reptilian eggs cannot be incubated underwater; amniote embryos in shelled eggs must exchange respiratory gases with the environment across the eggshell, and this exchange is much slower in water than in air[30]. Therefore, viviparity would have been highly adaptive for *Dinocephalosaurus* to reproduce in the sea. We also note that the sacral ribs are separated from the sacrum in *Dinocephalosaurus*[26], indicating a movable pelvis, a character that is interpreted as evidence for viviparity in numerous other Mesozoic marine reptiles[6,7].

Although archosauromorphs originated around 260 Ma in the Late Permian[18], the earliest evidence of reproductive biology in this group only came from the Early Jurassic dinosaurian embryos associated with calcified eggshells reported from South Africa[20] and China[21]. Thus, there is a gap of roughly 70 million years between the origin of archosauromorphs and the earliest evidence of reproductive biology in this group. Now the discovery of live birth in the Middle Triassic *Dinocephalosaurus* fills this gap and extends the previous understanding of reproductive biology in archosauromorphs by roughly 50 million years. Our discovery is also the only information available for the reproductive biology of the stem-group archosauromorphs.

Roughly one third of extant amniote species are archosauromorphs (mostly birds). Given the phylogenetic, morphological, and ecological diversity of extant archosauromorphs—birds in particular—the absence of viviparity in this group is striking, especially when compared with lepidosauromorphs (tuatara, lizards

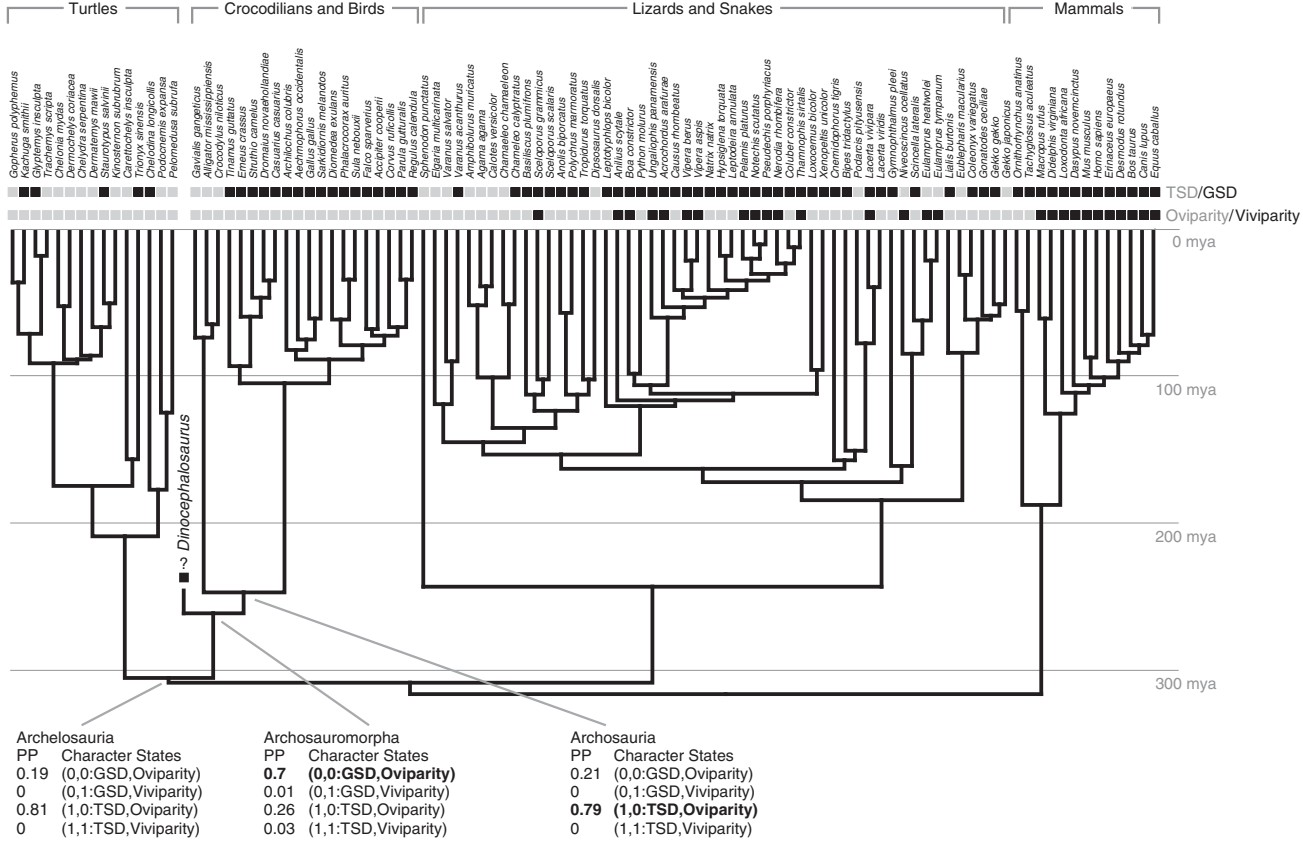

**Figure 5 | Distribution of sex determination mechanism and reproduction mode.** The stratogram summarizes all key amniote clades. GSD, genotypic sex determination; TSD, temperature sex determination. Posterior probabilities of ancestral states for Archelosauria, Archosauromorpha and Archosauria are given. The highest posterior probability is bolded.

and snakes). Multiple hypotheses have been put forward to explain the absence of viviparity in birds, including the biomechanical demands of flight, oviduct physiology and lack of the selection pressure to evolve viviparity[31,32]. Our discovery of a pregnant *Dinocephalosaurus* demonstrates that ancestrally there was no genetic or developmental impediment to evolve live birth in this diverse group. The reasons why live birth is absent in extant archosauromorphs may therefore be because of lineage specific constraints and adaptations rather than an attribute of the wider groups' underlying biology.

Sex determination in Archelosauria (turtles plus crocodilians and birds)[33] is diverse, with temperature-dependent sex determination in crocodilians and some turtles, the ZW genetic system in birds and some turtles, and the XY genetic system also in some turtles[34]. For nearly all amniotes, the evolution of live birth is dependent on the prior evolution of genotypic sex determination (GSD), with some skinks as the notable exception[35]. Phylogenetic modelling (Fig. 5) using this evolutionary relationship and the presence of live birth in the new specimen shows that *Dinocephalosaurus* likely had GSD (0.95 posterior probability). This suggests the presence of GSD in marine protorosaurs, consistent with the hypothesis that GSD and live birth were present in early diapsid lineages, and that both features are necessary to facilitate the land-water transition in lineages of obligate marine amniotes[35].

## Methods
**Phylogenetic analysis.** The phylogenetic relationship of *Dinocephalosaurus* with other protorosaurs has been assessed previously[26], which was derived from three

independently published data matrices[36–38]. The problem in Rieppel et al.'s[26] analysis is that many characters were repeated in the original sources, which resulted in an artificial weighting of different characters. There is a recent study assessing the phylogenetic relationship of protorosaurs, but *Dinocephalosaurus* was not incorporated into the new data matrix[39].

To assess the phylogenetic relationship of *Dinocephalosaurus* for the current study, we first constructed a data matrix by combining three published data matrices[36–38] and deleted the repeated characters. If a character is originally from Benton and Allen[36], Jalil[37] or Dilkes[38], it was noted as B/J/D plus the original character sequence in the relevant reference below. Compared with previous analysis, B3, B47, J1, D34, D84, D94 and D96 were difficult to follow and were deleted. D112 was deleted since it may easily be miscoded by slight postmortem alteration of the tarsal position. This resulted in 171 characters from the three previous analyses, some of which were slightly modified. Characters 13, 58, 103, 114, 122, 123, 124, 138 and 181 in this study are more or less new to the original data matrix of Rieppel et al.[26]. In addition, Characters 77 and 78 are from Modesto and Sues[40]. The revised data matrix used in our phylogenetic analyses includes 182 characters, of which 147 are parsimony informative.

The analysis is at the generic level. Unlike previous analyses[26], we did not include data from *Cosesaurus*, *Kadimakara* and *Trachelosaurus* because the material is poorly preserved and yields few characters useful for phylogenetic analysis. We also excluded *Malerisaurus* because its species may be chimaeras[41]. *Vallesaurus* was added to the new data matrix since it provides some information about the cranial morphology of drepanosaurs, which is lacking in the previous analysis[26]. This resulted in 20 taxa in total, where *Petrolacosaurus* was selected as the outgroup.

The data matrix (Supplementary Data 1) for the phylogenetic analysis was prepared using NDE Version 0.5.0. Details of the characters and the list of personally examined material are given in Supplementary Note 2 and Supplementary Table 1, respectively. PAUP Version 4.0 Beta 10 for Windows[42] was used to analyse the data matrix using the branch and bound algorithm with default settings to estimate the most parsimonious trees. All uninformative characters were excluded before the search. A single most parsimonious tree was obtained, with a tree length of 358, CI = 0.4609, HI = 0.5391, RI = 0.5768, RC = 0.2658. Details of statistical support of the tree can be seen in Supplementary Fig. 1.

We also used MrBayes v3.2.6 (ref. 43) to infer phylogenetic relationships within a Bayesian framework. Each analysis consisted of four Markov Chain Monte Carlo (MCMC)[44] chains run for 10,000,000 generations, with parameters sampled every 1,000 generations; the analysis was repeated four times. The average standard deviation of split frequencies between the MrBayes runs was <0.01, which indicates that the MCMC chains converged. We double-checked that the runs had reached a stationary phase by examining a time-series plot for the log-likelihoods in Tracer 1.6 (http://beast.bio.ed.ac.uk/Tracer), which showed that all parameters had effective sample size (ESS) >8,000 across both runs. The Bayesian phylogeny agrees broadly with the tree inferred using parsimony. Details of the Bayesian tree can be seen in Supplementary Fig. 2.

**Sex-determining mechanism.** We used discrete-coded data for the system of sex determination (0, genotypic versus 1, temperature-dependent and reproductive mode (0, oviparous versus 1, viviparous) for 101 extant amniote species from a previous study[35] plus *Dinocephalosaurus*. Although these traits sometimes show intermediate states, they generally vary discretely rather than continuously. The program BayesTraitsV2 (http://www.evolution.rdg.ac.uk/BayesTraits.html) was used to estimate the instantaneous rates of change among character states using a Markov Chain Monte Carlo procedure[44]. Markov chains were run for 5,100,000 iterations, sampling every 500 iterations, following a burn-in of 100,000 iterations.

Mesquite v3.03 (http://mesquiteproject.wikispaces.com/) was used to construct the phylogenetic tree following the previous study[35], with *Dinocephalosaurus* and turtles positioned as the consecutive sister groups to the archosaurs (Fig. 5). We also conducted an analysis that sampled two other phylogenies that varied only in the location of Testudines (turtles as the sister clade of Lepidosauria and Sauria respectively) to explore the uncertainty of phylogenetic position of Testudines within amniotes. The three phylogenies yielded identical results.

Each iteration was sampled over the three trees. State frequencies were estimated from the data. Following the best fitting model[35], we used hyperpriors that seed exponential rate priors from a flat distribution of 0–10 for all rates except q21, q24, q34 and q43, which were seeded using hyperpriors ranging from 0 to 0.1. The posterior rates of change and phylogenetic information (position and branch lengths) were used to infer the type of sex determination in *Dinocephalosaurus*. For discrete predictions, the frequency of a given state in the posterior distribution (the character state's posterior probability) is the degree of support for that character state. Ancestral character state estimation was also conducted for three nodes of interest (Archelosauria, Archosauromorpha and Archosauria). The results provide strong support that most recent common ancestors at these nodes were oviparous, though the mode of sex determination is equivocal.

**Data availability.** The authors declare that all data generated or analysed during this study are included in this published article and its Supplementary information files.

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

## Acknowledgements
J.L. would like to thank J. Ding, Y.Y. Sun and M.F. Zhou for facilitating this collaborative research in Chengdu, D.G. Blackburn, M. Sánchez-Villagra, R. Shine, J.R. Stewart and M.B. Thompson for discussion or providing literature on the evolution of viviparity in reptiles, and N. Bardet, C. Klug, C. Dal Sasso, J. Fortuny, J. Gallemí, À. Galobart, U. Göhlich, N.E. Jalil, D.Y. Jiang, C. Li, G. Muscio, A. Paganoni, O. Rieppel, T. Scheyer and L. Simonetto for providing access to specimens under their care. S.A. Gao collected the fossil specimen with the coordination of Q.Y. Zhang, C.Y. Zhou, S.X. Hu and J.Y. Huang in the field. H.Y. Wu prepared the fossil specimen in part. W. Wen identified the perleidid fish. D.C. Deeming, J.D. Gardner, R. Motani and P.M. Sander provided comments on the earlier versions of the manuscript. The research was funded by the National Natural Science Foundation of China (No. 41402015), State Key Laboratory of Palaeobiology and Stratigraphy (Nanjing Institute of Geology and Palaeontology, CAS) (No. 143104), the Natural Science Foundation of Anhui Province (No. 1508085QD70) and the Fundamental Research Funds for the Central Universities of China (No. 2014HGQC0026). Fieldwork and lab preparation were supported by China Geological Survey (Projects 12120114068001 and 1212010610211). This is part of a PhD project of J.L. under the supervision of J.C.A., M.F. Zhou and R. Motani.

## Author contributions
J.L. performed research, analysed data, prepared figures, and wrote the paper; C.L.O. conducted the Bayesian analysis and prepared relevant figures; C.L.O., M.J.B., M.C.B. and J.C.A. revised the paper; all authors discussed results and reviewed the paper.

## Additional information

**Competing financial interests:** The authors declare no competing financial interests.

