## [Peer Review File · Nature Communications]

Reviewers' comments:

Reviewer #1 (Remarks to the Author):

This paper reports on a highly significant discovery and associated analysis that are likely to be of broad interest to the scientific community. Although viviparity (live-bearing reproduction) has arisen independently in more than 150 separate lineages of vertebrates, no evidence for viviparity has ever been found among archosauromorphs – a very large group that includes birds, crocodylians, non-avian dinosaurs, and several other extinct groups).

The present study presents evidence for viviparity in a Triassic protosaur – representing the first report of this reproductive mode in any archosauromorph. Not only does this finding add another origin of viviparity to the growing list, but it counters past speculation that some biological feature has constrained its evolution in archosauromorph. The paper will thereby tend to shift the focus from assumptions about constraints on this pattern in archosaurs to consideration of the roles of selective pressures.

The evidence for viviparity in *Dinocephalosaurus* is based on discovery of a pregnant female containing a skeletonized fetus. The evidence appears to be strong, and far more plausible than alternative explanations (such as cannibalism and superimposition of fossil specimens). The detailed analysis convincingly places *Dinocephalosaurus* phylogenetically among protosaurs allied with tanystropheids. Likewise, application of phylogenetic models suggests that the animals had a genetic sex determination (rather than a temperature- dependent mechanism), a pattern thought to be a predisposing factor in other extinct marine amniotes to have evolved viviparity.

I can offer only two trivial suggestions for improvement.

The paper (line 41) indicates that evolution of viviparity “requires extensive modification of maternal morphology, physiology,” etc. To the contrary, numerous studies on squamate reptiles and fishes have shown that evolution of viviparity typically involves very small and subtle changes to these features. (More extensive modifications can accompany evolution of matrotrophy, but that is a different issue).

Line 122 states that eggs in all archosauromorphs and turtles are rigid – shelled. In fact, many chelonians have pliable- shelled eggs with a reduced level of calcification.

These small issues are easily addressed by small changes in wording, and should in no way delay publication of this first- rate paper.

Reviewer #2 (Remarks to the Author):

This paper describes an interesting new specimen of a Middle Triassic archosauromorph that clarifies reproductive mode in basal archosauromorphs. The paper is well written, and the

authors carefully lay out their evidence for the small individual being an embryo rather than stomach contents. The authors make a solid case that the specimen provides an example of live birth, a first for any archosauromorph known. This paper will be of interest to a broad audience interested in reproduction, the fossil record, and major trends in evolution.

However, there are a few points to address in the text:

Line 96-97: "They may be unossified, as consistent with the general ossification sequence of tetrapod limbs." This statement could use a reference.

Line 122: The authors state that all archosauromorphs and turtles have rigid-shelled eggs, but these are typically described as 'leathery'- are these not 'pliable'?

Line 133-134: "the bones of the embryo are well ossified" This seems contradictory to an earlier statement in line 96-97 where the authors discuss the lack of autopodia as potentially the result of ossification sequence, indicating the early ontogenetic stage of the embryo. Please clarify these two statements.

The figures leave something to be desired. Figure 1 is fine, though it appears grainy- this might just be low resolution for the submission. Figure 2 could use a star or other indicator of the specimen locality in part a, and in part b it would be good to include the geologic time along the left side along with Age, Formation, and beds. I find the photograph and line drawing of the specimen for Figure 3 to be entirely too small to see any details of the fossils. Go ahead and make it a full page- it's nearly at that size already. The orientations of c and e within the figure are slightly non-intuitive, and they would work better as a separate square component of the overall figure (in which they can be bigger) rather than being forced into the negative space around the specimen photograph. I don't understand the meaning of the colors in the cervical vertebrae of b because they don't seem to correspond to anything in the photograph or the figure caption. In the caption itself 'ul' should be 'ulna' rather than 'ulnare'. Figure 4 does not need to be this large. The clade names in the caption of Figure 4 could be put on the actual tree for clarity, leaving the caption space to be more informative.

The authors briefly address a recent study by Pritchard et al. (2015) on tanystropheids in their phylogenetic methods section, saying that *Dinocephalosaurus* wasn't included in that matrix. Why did the authors choose to not use that matrix and incorporate their taxon? It seems that they are recreating the potential problems that they discuss with repeated characters and possible artificial weighting using their current method of combining three data matrices to assess the phylogenetic position of *Dinocephalosaurus*. Why was a branch-and-bound algorithm chosen rather than performing a heuristic search?

Extended Data Figure 3 is also quite small and difficult to read. This would work better oriented left to right rather than oriented up and down.

In the Supplementary Information the authors include a "Detailed description and comparison", however, there are almost no comparisons made. These should be included here. In Line 519 it should be 'phalanx' rather than 'phalange'. The 'examined specimens and referred literature' does not include several new citations that would be relevant to this

study, namely Nesbitt et al. (2016) on the postcranium of Azendohsaurus with information on Trilophosaurus and Pritchard et al. (2015) and (2016) on tanystropheids and drepanosaurs, respectively.

Reviewer #1 (Remarks to the Author):

1. The paper (line 41) indicates that evolution of viviparity “requires extensive modification of maternal morphology, physiology,” etc. To the contrary, numerous studies on squamate reptiles and fishes have shown that evolution of viviparity typically involves very small and subtle changes to these features. (More extensive modifications can accompany evolution of matrotrophy, but that is a different issue).

Reply: we agree with the referee and changed the text to read “The evolutionary transition from egg laying (oviparity) to live birth (viviparity) involves subtle changes to maternal morphology, physiology, and behavior, which can lead to matrotrophy and shift ecological and evolutionary trajectories.”

2. Line 122 states that eggs in all archosauromorphs and turtles are rigid – shelled. In fact, many chelonians have pliable- shelled eggs with a reduced level of calcification.

Reply: we have changed to “In all archosauromorphs and turtles, the eggshells are well-calcified, though many turtle eggs are also pliable” and cited a new reference (Deeming and Ferguson, 1991) to support this statement.

Reviewer #2 (Remarks to the Author):

1. Line 96-97: “They may be unossified, as consistent with the general ossification sequence of tetrapod limbs.” This statement could use a reference.

Reply: as suggested, a reference (Benton, 2014) has been cited to support this statement.

2. Line 122: The authors state that all archosauromorphs and turtles have rigid-shelled eggs, but these are typically described as ‘leathery’- are these not ‘pliable’?

Reply: see the reply #2 to Reviewer 1 above.

3. 133-134: “the bones of the embryo are well ossified” This seems contradictory to an earlier statement in line 96-97 where the authors discuss the lack of autopodia as potentially the result of ossification sequence, indicating the early ontogenetic stage of the embryo. Please clarify these two statements.

Reply: in the original line 96-97, the lack of autopodia indicates that the embryo is an embryo instead a fully mature adult. In the original line 133-134, when we say that “the bones of the embryo are well ossified”, we indicate that the embryo is in “a relatively advanced *embryonic* stage”.

4. Figure 2 could use a star or other indicator of the specimen locality in part a, and in part b it would be good to include the geologic time along the left side along with Age, Formation, and beds.

Reply: a star and geologic time have been added into Fig. 2a and 2b respectively, as suggested.

5. I find the photograph and line drawing of the specimen for Figure 3 to be entirely

too small to see any details of the fossils. Go ahead and make it a full page- it's nearly at that size already. The orientations of c and e within the figure are slightly non-intuitive, and they would work better as a separate square component of the overall figure (in which they can be bigger) rather than being forced into the negative space around the specimen photograph. I don't understand the meaning of the colors in the cervical vertebrae of b because they don't seem to correspond to anything in the photograph or the figure caption. In the caption itself 'ul' should be 'ulna' rather than 'ulnare'.

Reply: These are good suggestions. We have now formatted the Fig. 3 into a full page following the journal's requirement. We prefer to keep the original orientations of panels c and e since once we reformatted these two panels following the referee's suggestion, there will be plenty of empty space in the original figure. In addition, adding these two new panels will make the other panels of the figure much smaller and much less clear. Now we have uploaded much higher-resolution figure so the details will be able to be seen when zoom out. The different color in the cervical region aims to facilitate the association of cervical ribs with corresponding vertebrae. We have now added this explanation in the revised manuscript. "Ulnare" has been changed to "ulna" as suggested.

6. Figure 4 does not need to be this large. The clade names in the caption of Figure 4 could be put on the actual tree for clarity, leaving the caption space to be more informative.

Reply: we have now formatted the Fig. 4 following the journal's requirement. We prefer to keep the clade names in the legend since adding these clade names into the figure will make it crowded.

7. The authors briefly address a recent study by Pritchard et al. (2015) on tanystropheids in their phylogenetic methods section, saying that *Dinocephalosaurus* wasn't included in that matrix. Why did the authors choose to not use that matrix and incorporate their taxon? It seems that they are recreating the potential problems that they discuss with repeated characters and possible artificial weighting using their current method of combining three data matrices to assess the phylogenetic position of *Dinocephalosaurus*. Why was a branch-and-bound algorithm chosen rather than performing a heuristic search?

Reply: the goal of our phylogenetic analysis was only to find whether *Dinocephalosaurus* was an archosauromorph. Incorporating Pritchard et al. (2015)'s data matrix into our analysis won't affect the conclusions of our paper, which focus on the evolution of live birth and not detailed taxonomy.

The three independent data matrixes by Benton and Allen (1997), Jalil (1997) and Dilkes (1998) use many similar or identical characters. Rieppel et al's (2001, 2008) study simply combined these characters into a data matrix without the deletion of repeated characters, resulting in uneven character weighting. We have redefined some characters and deleted repeated characters originally used by Rieppel et al. (2001, 2008), as we explained in the Methods section. This

avoids the artificial character weighting. We also expanded the original data matrix by adding new characters via the comparative study of protorosaurs.

The branch-and-bound algorithm can search the globally most parsimonious trees when compared with heuristic search. Only “*when a data set is too large to permit the use of exact methods (such as branch-and-bound algorithm), we must resort to heuristic approaches that sacrifice the guarantee of optimality in favor of reduced computing time*” (Swofford, 1993). Further explanations can be found in standard reference books (e.g., Swofford, 1993, PAUP; Felsenstein, 2004, *Inferring Phylogenies*, published by Sinauer Associates).

8. Extended Data Figure 3 is also quite small and difficult to read. This would work be better oriented left to right rather than oriented up and down.

Reply: we have re-formatted the figure and moved it from supplementary information to the main text.

9. In the Supplementary Information the authors include a “Detailed description and comparison”, however, there are almost no comparisons made. These should be included here.

Reply: the comparison was performed with the published *Dinocephalosaurus* specimens whenever possible. This is our original meaning of comparison. To avoid confusion, we have slightly redrafted the subheading. For the comparison of *Dinocephalosaurus* with other taxa, readers can easily find the morphological difference in our morphological data matrix.

10. In Line 519 it should be ‘phalanx’ rather than ‘phalange’. The ‘examined specimens and referred literature’ does not include several new citations that would be relevant to this study, namely Nesbitt et al. (2016) on the postcranium of *Azendohsaurus* with information on *Trilophosaurus* and Pritchard et al. (2015) and (2016) on tanystropheids and drepanosaurs, respectively.

Reply: all suggestions accepted and revised accordingly.

REVIEWERS' COMMENTS:

Reviewer #2 (Remarks to the Author):

The authors appear to have addressed all comments by both reviewers, specifically updating figures as requested and clarifying text. I have no further comments on this manuscript and look forward to seeing this interesting work in publication!